**Development of a dynamic dust-source map for NMME-DREAM v1.0 model based on MODIS NDVI over the Arabian Peninsula**

Solomos Stavros[1,2], Abdelgadir Abuelgasim[2*], Christos Spyrou[3], Ioannis Binietoglou[4], Slobodan Nickovic[5]

**Abstract** We developed a time dependent dust source map for NMME-DREAM v1.0 model based on the satellite MODIS Normalized Difference Vegetation Index (NDVI). Areas with NDVI<0.1 are classified as active dust sources. The updated modeling system is tested for dust emission capabilities over SW Asia using a mesoscale model grid increment of 0.1°×0.1° km for a period of one year (2016). Our results indicate significant deviations in simulated Aerosol Optical Depths compared to the static dust-source approach and general increase in dustloads over the selected domain. Comparison with MODIS Aerosol Optical Depth (AOD) indicates a more realistic spatial distribution of dust in the dynamic source simulations compared to the static dust sources approach. The modeled AOD bias is improved from -0.140 to 0.083 for the case of dust events (i.e. for AOD >0.25) and from -0.933 to -0.424 for dust episodes with AOD>1. This new development can be easily applied to other time periods, models and different areas worldwide for a local fine tuning of the parameterization and assessment of its performance.

[1] Institute for Astronomy, Astrophysics, Space Applications and Remote Sensing (IAASARS), National Observatory of Athens, Athens, Greece, stavros@noa.gr

[2] Department of Geography and Urban Planning, National Space Science and Technology Center, United Arab Emirates University

[3] Department of Geography, Harokopio University of Athens (HUA), El. Venizelou Str. 70, 17671 Athens, Greece.

[4] National Institute of R & D for Optoelectronics, Magurele, Ilfov, Romania

[5] Republic Hydrometeorological Service of Serbia , Belgrade, Serbia

*Corresponding author

Keywords: dust, Arabian Peninsula, DREAM, NDVI, model, satellite

**Introduction**

The importance of natural particles, namely desert dust, in the weather and climate has been underlined in a great number of studies. Dust is a climatic regulator, as it modifies extensively the radiative balance of the atmospheric column (e.g. Torge et al., 2011; Spyrou et al., 2013; Mahowald et al., 2014). At the same time dust aerosols modify the atmospheric water content (Spyrou 2018), the way clouds are formed by acting as cloud condensation nuclei (CCN) and ice nuclei (IN) and the precipitation process (Kumar et al., 2011; Solomos et al., 2011; Nickovic et al., 2016). In addition, there is a clear connection between dust particles and human health disorders, as the size of the produced aerosols is small enough to cause respiratory and cardiovascular diseases, as well as pathogenic conditions due to the microorganisms that they can potentially carry (Mitsakou et al., 2008; Esmaeil et al., 2014).

The Arabian Peninsula is one of the most important sources of mineral dust worldwide and contributes together with the Saharan and Gobi Deserts in the formation of a North Hemisphere "dust belt" as described by Prospero et al. (2002). Severe dust storms over the Peninsula are quite common, especially during long periods without rain, in the spring and summer (Almazrouia et al., 2012). Particles injected into the atmosphere from arid soils, under favorable weather conditions (high wind speeds and dry soil), can affect large areasaround the sources but also remote locations like the Eastern Mediterranean (Mamouri et al., 2016; Solomos et al., 2017) and the Indian Ocean (Chakraborty et al. 2006).

Due to the multitude and severe effects of dust particles not only on the weather and the ecosystem but to human health as well, the proper description of the production, transport and eventual deposition of the dust cycle, in numerical weather prediction models (NWPs) is essential. In order to be able to accurately describe the dust life-cycle in the atmosphere, we need a clear understanding of the areas which can potentially act as "dust sources". The definition of such areas dictates the emission strength and therefore the amount of particles inserted into the atmosphere. A proper representation of dust sources is therefore an essential first step, in studying the impacts of mineral particles in the climate and human societies. Usually the definition of the areas that can act as dust sources is made using global datasets. For example Nickovic et al. (2001) used a subjective correspondence between the Olson World Ecosystems (Olson et al., 1983) and the thirteen SSib (simplified simple biosphere, Xue et al. 1991) vegetation types to identify arid and semi-arid areas. Similarly, Spyrou et al., (2010) used a 30sec global land use/cover database, classified according to the 24 category U.S. Geological Survey (USGS) land use/cover system (Anderson et al., 1976), to define active areas in SKIRON dust model. Solomos et al., (2011) used the LEAF soil and vegetation sub-model of the Regional Atmospheric Modeling System (RAMS) (Walko et al., 2000) to identify the active dust sources in RAMS-ICLAMS model.

However, the above mentioned methodologies have some significant drawbacks.The datasets are usually not up-to-date, therefore recent land-use modifications are not included and not represented. In addition, such "static" databases mean that possible seasonal variations are not taken into account. Towards the direction of overcoming the above limitations and improving global dust forecasts, Kim et al., (2013) developed a dynamical dust source map for the GOCART dust model by characterizing NDVI values < 0.15 as active dust spots. Similarly Vukovic et al., (2014) combined MODIS landcover types with pixels having NDVI < 0.1 to identify the seasonal dust sources that enforced the severe Phoenix haboob of July 2011 in the US. Such information can be even more relevant at meso and local scales for determining landuse changes and potential dust sources, especially in heterogeneous regions such as the Arabian Peninsula (which has more diverse soil types than e.g. the Sahara Desert) and the greater SW Asia. In this context, Solomos et al., (2017), used the Landsat-8 NDVI data (assuming also NDVI<0.1 as active sources) to identify recent changes in landuse due to the war in Iraq and Syria resulting in a significantly more realistic simulation of dust properties in the Middle East.

In the current study we present the implementation of a dynamical dust source map in the well-established and widely used DREAM v1.0 dust model (Nickovic et al., 2001; Perez et al., 2006). The new development is first tested here for the greater SW Asia but can be extended for use in mesoscale dust modeling applications worldwide. Two experimental simulations are

performed for one month period (August 2016) over the greater SW Asia: 1) Control run, where
the dust source definition is based on the Ginoux et al., (2001) dataset and 2) Dynamic source
run, where the NDVI values are used to identify the dust sources. The main differences in our
approach compared to the previous studies referenced above, is that we use a very high
resolution NDVI product (500×500 m) in a regional modeling domain (e.g. Kim et al., 2013 used
an 8×8 Km NDVI dataset extrapolated to 1°x1° global modeling domain) and our study is not
limited to specific test cases (like for example Vukovic et al., 2014 and  Solomos et al., 2017),
but covers an extended time period, as presented below. The model results from both runs are
compared to available satellite observations and station measurements inside the modeling
domain. In section 1 we describe the methodological steps regarding the model developments
and remote sensing data; Section 2 includes the results of the experimental runs and section 3
is a summary and discussion of the study findings.

## 1. Methodology

### 1.1. Model description
The modeling system used in this study is NMME-DREAM v1.0. The meteorological core is
the NCEP/NMME atmospheric model (Janjic et al., 2001). The Dust Regional Atmospheric Model
(DREAM v1.0) is a numerical model created with the main purpose to simulate and predict the
atmospheric life-cycle of mineral dust using an Euler-type nonlinear partial differential equation
for dust mass continuity (Nickovic et al., 2001; Perez et al., 2006; Pejanovic et al., 2011, Nickovic
et al., 2016). In DREAM the concentration approach is used for dust uplift, where surface
concentration is used as a lower boundary condition and used for the calculation of surface
fluxes, which in turn depends of the friction velocity (Nickovic et al., 2001). This surface
concentration is calculated using equation (11) from Nickovic et al., (2001):

$$C_{sfc} = c_1 \cdot \delta \cdot u_*^2 \left[ 1 - \left( \frac{u_{*t}}{u_*} \right)^2 \right]$$ where $c_1 = 2.4 \cdot 10^{-4} \frac{Kgr}{m^5 \sec^2}$ a constant determined from model
experiments, $u_*$ and $u_{*t}$ the friction velocity and the threshold friction velocity for dust
production respectively and $\delta = a \cdot \gamma_k \cdot \beta_k$ , where $\gamma_k$ the ratio between the mass available for
uplift and the total mass $\beta_k$ the fractions of clay, silt and sand for each soil class, and $a$ the
desert mask (between 0 and 1) calculated from the Ginoux et al., (2001) dataset. Soil moisture
and particle size dictate the threshold friction velocity which initializes dust production. Once
particles have been lifted from the ground they are driven by the atmospheric model variables
and processes. Therefore turbulent parameters are used in the beginning of the process, when
dust is lifted from the ground, and transported by model winds in the later phases when dust
travels away from the sources. The model handles dust in eight size bins, with effective radii of
0.15, 0.25, 0.45, 0.78, 1.3, 2.2, 3.8, and 7.1 mm. Dust is treated as a passive tracer and doesn`t
interact with radiation or clouds. Dust is eventually settled through rainfall and/or dry
deposition processes parameterized according to the scheme of Georgi (1986) which includes
deposition by surface turbulent and Brownian diffusion, gravitational settling and impact on
surface elements.
In order to test the use of NDVI for source characterization, the model is setup with a horizontal
resolution of 0.1°x0.1°, covering the Arabian Peninsula parts of SW Asia and parts of NE Africa
(Figure 1). On the vertical we use 28 levels stretching from the surface to the top of the
atmosphere. August 2016 has been selected as a test period for the model development due to
the significant dust activity and variability in wind properties during this month. One-year runs
for the entire 2016 have been conducted to evaluate the performance of the static and
dynamic database emission maps .The original classification of dust sources in DREAM is based
on Ginoux et al., (2001) that takes into account the preferential sources related to topographic
depressions and paleolake sediments. The global mapping of dust sources in Ginoux et
al.,(2001) is determined from the comparison between the elevation of surface grid points at
1°×1° resolution with the surrounding hydrological basins and with the 1°×1° AVHRR (Advanced
Very High Resolution Radiometer) vegetation map (DeFries and Townshend, 1994). Recent
studies indicated the contribution of both natural and anthropogenic dust sources to the overall
dust emissions detected in MODIS Deep Blue product (Ginoux et al., 2012) and also the
relevance of local geomorphological conditions and sediment supply (Parajuli and Zender,
2017) on the global dust emissions. All these advances in dust emissions are based on static
map considerations.
In our work, a numerical procedure has been developed to insert the NDVI satellite information
into the model and to update such info each time the NDVI changes, during the simulation
period. We assume that regions with NDVI values from 0 to 0.1 correspond to bare soil and
therefore can be efficient sources ("dust points"; DeFries and Townshend, 1994; Solomos et al.,
2017). In general it is not easy to define a global threshold value for all satellite NDVI sensors
and all vegetation types worldwide.  For example Kim et al. (2013) used a threshold of 0.15 to
define global dust sources based on AVHRR retrievals (Tucker et al., 2005; Brown et al., 2006).
Here we adopt the 0.1 NDVI threshold due to the bareness of the specific modeling domain
since a higher value could overestimate the regional dust sources. The NDVI dataset is at finer
resolution than the model grid (500×500 m) and in order to find the potential for dust
production in each model grid box, we calculate the following ratio:

$$A_{grid\_box} = \frac{\#\_of\_dust\_points}{Total\_\#\_of\_points}$$

Where $\#\_of\_dust\_points$ is the number of points with NDVI values smaller than 0.1.This
approach allows for a dynamic description of dust source areas over the model domain to
replace the previously used static database. Moreover, the scaling of satellite data over model
grid points allows the use of the same algorithm for different model configurations. Several
mountains in the area (e.g. the Sarawat Mountains along the Red Sea coast and the Zagros
Mountains in Iraq) could be misclassified as dust sources due to low NDVI values.  In order to
exclude such unrealistic emissions from non-soil bare areas or snow-covered areas we have
applied a limit of zero dust production above 2500 m over the entire domain. This simple
approach has been selected in order to keep our straightforward NDVI mapping independent of
vegetation and soil information. The threshold value of 2500 m does not suppress the
emissions from lowlands and hillsides (e.g. the coastal areas of Hejaz Mountains in Red Sea that
have been identified as hot dust spots by Anisimov et al., 2017).
In Figure 2a we show the static sources in the original model version with a factor of 0 to 1
depending on the source area strength. Accordingly in Figure 2b we show the new dynamic
sources for 1-16 of August 2016. The two dust source patterns present remarkable difference
especially over the western Saudi Arabia and over Iran and Pakistan where the NDVI
classification results in stronger emissions. In order to test the performance of the new
methodology we run the model in two different configurations: (1) Using the static Ginoux et
al., (2001) dust source database, called DREAM-CTRL run from now on, and (2) using the
dynamic NDVI database as described above, called DREAM-NDVI run from now on. Both setups
are initialized using the NCEP GFS analysis files (0.5°×0.5° at 00, 06, 12 and 18 UTC), which were
used for boundary conditions as well. The two model configurations are identical other than the
dust source database.

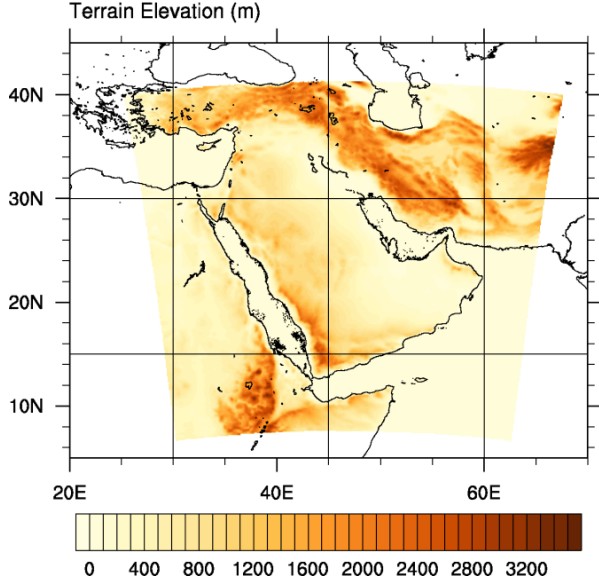


**Figure 1: DREAM model domain and topography in meters**
**1.2 NDVI description**
For the purposes of our study we used the 500m 16-day averaged NDVI from MODIS (Didan,
2015) for the period of interest. The NDVI is a normalized transform of the near infrared to red
reflectance ratio, designed to provide a standard for vegetation and takes values between -1
and +1. Since it is expressed as a ratio, the NDVI has the advantage of minimizing certain types
of band-correlated noise (positively-correlated) and influences attributed to variations in
irradiance, clouds, atmospheric attenuation and other parameters (Solano et al., 2010).
To create an accurate time-dependent dust source map, we have utilized the Normalized
Difference Vegetation Index (NDVI) derived from the MODIS/Terra instrument. NDVI is
calculated as the normalized difference of reflectance in the red and near-infrared channels
(Rouse et al., 1974; Huete et al. 2002) i.e.,

$$NDVI = \frac{X_{nir} - X_{red}}{X_{nir} + X_{red}}$$

where X represents surface reflectance as would be measured at ground level (i.e. corrected for atmospheric gas and aerosol effects).in each channel. The 16-day composite is calculated by ingesting two 8-day composite surface reflectance granules, taking into account pixel quality, presence of clouds, and viewing geometry. This procedure can lead to spatial discontinuities, as it is possible that data from different days are used for adjacent pixels, each representing different measurement conditions. If a pixel had no useful measurements during the 16-day period, historic data are used as fill values (Didan et al., 2015). For terrestrial targets, NDVI will take values near 0.8 for vegetated areas and near 0 for barren soil (Huete et al., 1999). The high-resolution dataset was used to calculate the percentage of barren land in each 0.1°x0.1° model grid cells and this percentage was used to define the effective strength of dust sources in each cell.

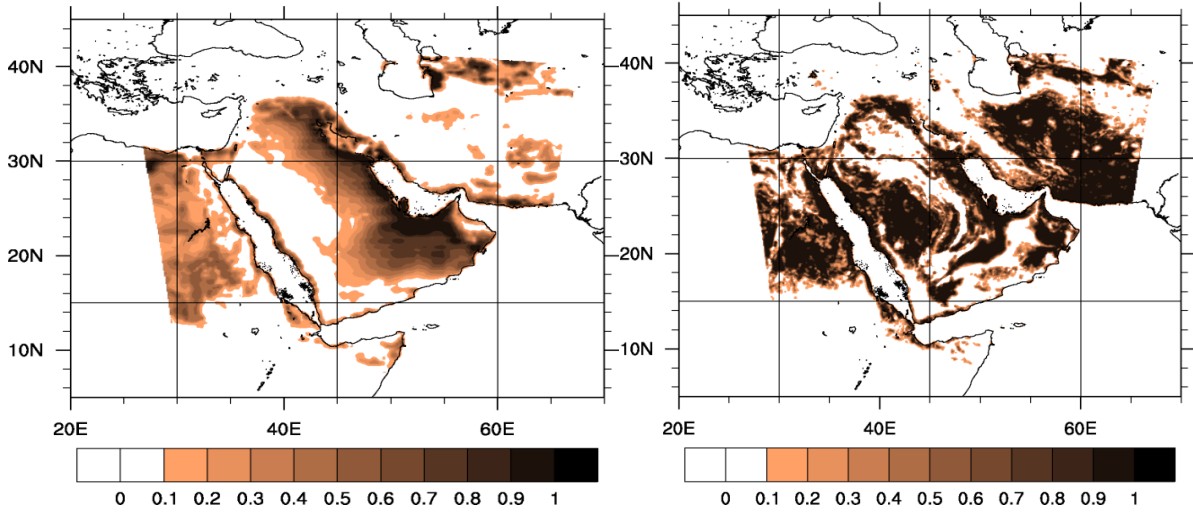

**Figure 2: Dust source strength as defined by (a) the Ginoux et al., 2001 dataset and (b) the 1-16th of August 2016 mean NDVI**

**1.3 Evaluation datasets and metrics**

Model evaluation is carried out two datasets. First, the MODIS monthly aerosol optical depth (AOD) is use to study the spatial distribution of dust in the model domain. For this we use the level 3 gridded atmosphere monthly product at 1x1 resolution, MOD08_ME (Platnick et al. 2017). Secondly, we evaluate model performance using AERONET AOD retrievals at 8 photometeric stations. AERONET is a network of sun/sky photometers that derive aerosol optical and microphysical properties at a large number of stations around the world (Holben et al., 1998). For this evaluation, we use Version 3 AOD retrievals that, in comparison with previous versions, improves automatic cloud screening (Giles et al, 2018). Level 2 datasets were used for all stations apart from Kuweit University, where only Level 1.5 data were available. Both model and AERONET AOD were calculated at 532nm; this was chosen to facilitate future intercomparing against lidar systems that frequently measure at this wavelength (e.g.

Pappalardo et al., 2014). AERONET measurements were converted to this wavelength using the
440-870 angstrom exponent and taking into account AOD measurements at 440nm, 675nm,
and 870nm; in the cases where the 440nm AOD was not available, the 500nm (Mezaira) or
443nm (KAUST campus) measurement was used instead.

We evaluation model performance using five metrics: mean bias, root mean square error,
correlation coefficient, mean fractional bias, and fractional gross error. Concretely, assuming
we have n pairs of model values ($m_i$) and observations ($o_i$), the mean bias (MB) is defined as:
$$MB = \overline{m_\iota - o_\iota}$$

where the bar denotes the mean value. Root mean square error (RMSE) is defined as
$$RMSE = \sqrt{\overline{(m_\iota - o_\iota)^2}}$$

The correlation coefficient (r) is defined as
$$r = \frac{\sum_{i=1}^{n}(m_i - \bar{m})(o_i - \bar{o})}{\sqrt{\sum_{i=1}^{n}(m_i - \bar{m})^2}\sqrt{\sum_{i=1}^{n}(o_i - \bar{o})^2}}$$

The fractional gross error (FGE) is defined as
$$FGE = 2 \left| \overline{\frac{m_\iota - o_\iota}{m_\iota + o_\iota}} \right|$$

following Boylan and Russell, 2006. Similarly, mean fractional bias (MFB) is defined as
$$MFB = 2 \frac{\overline{m_\iota} - \overline{o_\iota}}{\overline{m_\iota} + \overline{o_\iota}}$$



following Chang and Hanna, 2004.


**2. Results**
DREAM-CTRL runDREAM-NDVI runThe test simulation period is 1-31 August 2016 and the
results from both simulations are compared to MODIS and AERONET AOD. A five days spin up
model run, prior to the experimental period, is used for establishing the dust background over
the domain. After finalizing the experimental model configuration we perform a complete one-
year run (2016) and evaluate the results against AERONET stations.
**2.1 Dust transport during August 2016**
The selected 1-month period is characterized by a significant variability in wind speeds and
directions (Figure 3) which allows the evaluation of the new model version under different
conditions. During 1-10 August, east winds prevail over the region and increased dust
concentrations are found mostly along the central, east and south coastal areas of the Arabian
Peninsula. An anticyclonic circulation is established during 10-15 over the Arabia Desert and
increased dust concentrations are mostly found over the central desert areas. On 16-26 August
the circulation is mainly from north directions and thick dust plumes are advected southwards
towards the Arabian Sea. The north winds veer to east on 26-31 August and increased
dustloads are found over the Gulf during these dates.

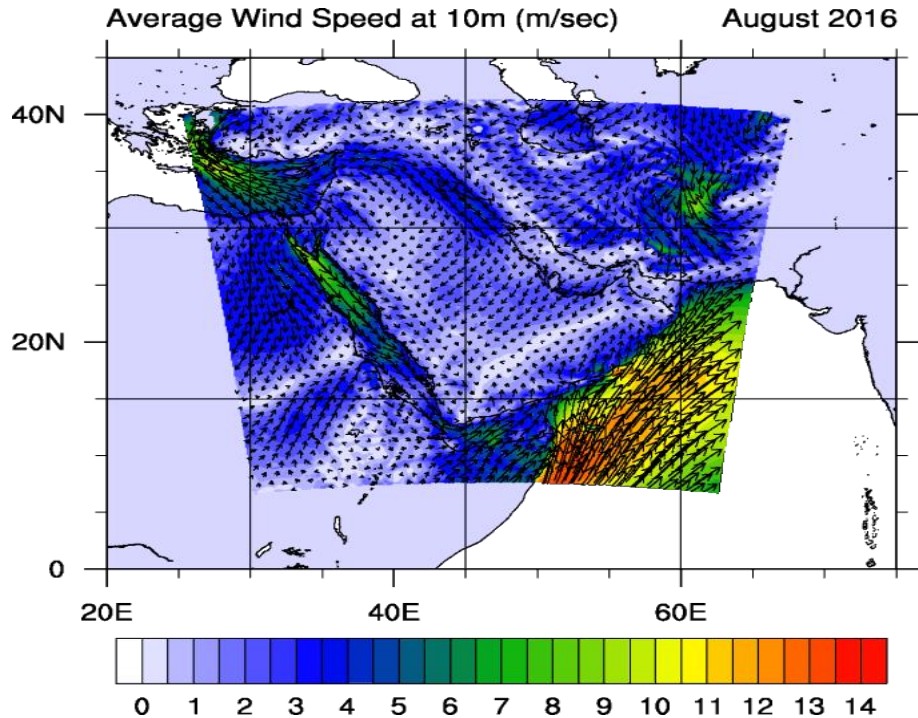

Figure 3. Average wind speed (color scale) and vectors from NMME-DREAMv1.0 for August
260  2016.

**2.2 Comparison with MODIS and AERONET**
The monthly average AOD for August 2016 is shown in Figure 4 for the two experimental runs
(Figure 4a,b). The DREAM-NDVI run results in a significantly modified spatial distribution of dust
presenting increased dustloads over the entire domain and most profoundly over the Red Sea
and Gulf regions (Figure 4b). This dust pattern is closer to the MODIS observed AOD over the
same period that is shown in Figure 4c. The MODIS AOD in this area is mostly related to dust,
however it must be taken into account that other aerosols not parameterized in the model (e.g.
sea salt, sulphates, nitrates) may also contribute to the observed MODIS AOD.
The first step is to examine how our methodology compares against the monthly average AOD
in our study area. Therefore the monthly average AOD values produced from our two
simulations (DREAM-NDVI run and DREAM-CTRL run) are compared. More specifically the
DREAM-NDVI run reproduces the MODIS observed AOD pattern that is in general characterized
by values 0.3-0.4 at the NW parts of the Arabian Peninsula and by values 0.4-0.8 at the SE parts.
Significant improvement is also evident over the Red Sea and NE Africa. The DREAM-NDVI run
captures the maximum observed AOD values reaching up to 1.6 over the Red Sea and also the
southwesterly extension of an AOD tongue of 0.3-0.8 towards Soudan.  At the east parts of the
modeling domain the DREAM-NDVI run again outperforms the DREAM-CTRL run since it
reproduces the spatial distribution of AOD 0.4-0.8 over the Arabian Sea and the maximum of
0.8-1.2 at the SE edge of Arabian Peninsula. Inside the Gulf, the NDVI run correctly represents
the 0.4-0.8 AOD but the dust concentration is over-predicted at the Strait of Hormuz and along
the Iran - Pakistan coastline. This is mostly due to the prevailing NE winds during the last days of
the August 2016 modeling period and due to a possible miss-classification of Iran and Pakistan
grid points as effective dust sources thus favoring unrealistic southeasterly transport towards
the Gulf of Oman.  The DREAM-NDVI AOD is also higher than MODIS AOD over western Saudi
Arabia indicating a possible overprediction of dust sources at this area.
As a second step we run the same model configurations (CTRL and NDVI) for the entire 2016.
The modeled dust optical depth is compared with individual AERONET measurements. The
model retrievals are interpolated in time to match the AERONET measurement time
considering only dust relevant measurements with Angström Coefficient <0.6 (Holben et al.,
1998) and the results are shown in Table 1. For completeness we first consider all AERONET
stations inside the modeling domain for the evaluation. However the stations that are at the
margins of our domain (Cairo_EMA_2, SEDE_BOKER, AgiaMarina_Xyliatou and El_Farafra) are
also affected by other dust source areas (e.g. Sahara Desert) and their statistics are not
representative for Arabian and Middle East sources. Instead, the comparison with Arabian
Peninsula stations (Eilat, Kuwait_University, KAUST_Campus and Mezaira) provides more
insight on the effects of the new source characterization. As seen in Figure 5 and also in Table 2
these stations are clearly benefited from the experimental run.
In general the two runs present a significant statistical difference and more remarkably a
reverse of bias (MODEL-AERONET) from negative in the DREAM-CTRL run to positive in the
DREAM-NDVI run. The DREAM-NDVI run produces increased AODs that are neither linearly
proportional to the DREAM-CTRL run AODs nor uniformly distributed over the domain. When
considering only Arabian stations, the statistical metrics in Table 1 and especially the fractional
gross error and bias are improved but the RMSE is increased due to the increase in maximum
modeled AODs. In order to investigate the sensitivity of our results towards the severity of dust
events we further assume two additional air quality states in Table 1: (i) dust events (AOD>0.25)
and (ii) severe dust episodes (AOD>1). Both cases show an improvement in the bias values over
the control simulations. When we consider AOD>1 the DREAM-NDVI run still underestimates
the observed values, but with a lower RMSE (0.586 versus 0.983 of the DREAM-CTRL run).This is
clearly evident in Figure 6 where the NDVI run is indeed more realistic for the Arabian stations
but still does not reproduce the extreme AOD during severe episodes. For most of the cases
such high AODs should be attributed to duststorms from convective downdrafts (haboobs).
These processes are not resolved at mesoscale model resolutions (Solomos et al., 2012, 2017;
Vukovic et al., 2014) and thus cannot be represented here.

**Table 1. Statistical metrics from the comparison between the annual runs and AERONET**

| | Mean bias (Model-Observation) | | RMSE | | Correlation | | Fractional gross error | | Mean fractional bias | |
|---|---|---|---|---|---|---|---|---|---|---|
| | CTRL | NDVI | CTRL | NDVI | CTRL | NDVI | CTRL | NDVI | CTRL | NDVI |
| AOD > 0 (All Stations) | -0.163 | 0.015 | 0.258 | 0.312 | 0.408 | 0.464 | 0.887 | 0.803 | -0.639 | 0.043 |
| AOD > 0 (Arabia Stations) | -0.142 | 0.122 | 0.252 | 0.332 | 0.340 | 0.426 | 0.644 | 0.515 | -0.455 | -0.187 |
| AOD > 0.25 ( Arabia Stations ) | -0.140 | 0.083 | 0.283 | 0.350 | 0.238 | 0.328 | 0.640 | 0.462 | -0.527 | -0.142 |
| AOD > 1 ( Arabia Stations ) | -0.933 | -0.424 | 0.983 | 0.586 | 0.032 | 0.009 | 1.230 | 0.481 | -1.211 | -0.413 |

The AERONET stations used in this study are: Eilat (29N,34E), Cairo_EMA_2 (30N,31E), Kuwait_University (29N,47E), KAUST_Campus (22N,39E), SEDE_BOKER (30N,34E), AgiaMarina_Xyliatou (35N,33E), Mezaira (23N,53E) and El_Farafra (27N,27E)

## 3. Summary and Discussion

In this study we present the development of a dynamic dust source map for implementation in NMME-DREAM v1.0 over the Arabian Peninsula and the greater areas of Middle East, SW Asia and NE Africa. Although the major dust sources worldwide are located in permanent deserts where the NDVI is almost always <0.1 (e.g. Bodele Depression, Gobi Desert, Arabian Desert), the dynamical scaling of dust emissions presented here can be important for providing up-to-date evidence of active dust sources over non-permanent deserts. These may include dried bog, marshes and semi-desert areas as well as irrigated and non-irrigated farms where landuse changes occur throughout the year. Analysis of the modeling results for one year test period (2016) over SW Asia indicated the improved performance of the new parameterization. The DREAM-NDVI run showed a significant increase in dustloads over the greater Arabian Peninsula area and a more realistic representation of the spatial distribution of AOD compared to the corresponding MODIS satellite retrievals. These findings support the previous results by Kim et al., 2013 who also showed an increase in dust emissions and a more realistic comparison with satellite observations in Saudi Arabia by the introduction of an NDVI based dynamic source mapping for GOCART model. Comparison with AERONET measurements also showed significant improvement especially at higher AODs that are also relevant to the model efficiency for air quality purposes (i.e. the model bias is reduced from -0.140 to 0.083 at AOD>0.25 and from -0.933 to -0.424 at AOD>1). However, the model statistics are not improved for all AERONET measuring stations and for all air quality states (Table2), mainly due to a possible misclassification of dust sources in the highlands of Iran and Pakistan.

The main purpose of our work was the development and first testing of this new modeling version. A major advance of our study is the ability to implement the real-time properties of dust sources in air quality simulations (as represented by the satellite NDVI) and thus capture

local or seasonal effects. In general, one year is not sufficient for extracting robust statistical
results and further analysis is required to examine the performance of the proposed
methodology over longer time periods and also over different areas worldwide. For example
the simple approach of employing a uniform value of NDVI<0.1 for determining the active dust
sources may not be adequate to represent fine-scale land properties and further adjustments
may be required depending on local-scale characteristics.  This new approach for the dynamic
characterization of active dust sources based on NDVI can be easily implemented in other
atmospheric dust models at different configurations and spatial coverage for improving their
performance.
**Table 2. Statistical metrics from the annual runs (2016) at AERONET stations. Bold values indicate correlation**
**coefficient with p <0.01.**

| Station | Mean bias | | RMSE | | Correlation | | Fractional gross error | | Mean fractional bias | |
|---|---|---|---|---|---|---|---|---|---|---|
| | CTRL | NDVI | CTRL | NDVI | CTRL | NDVI | CTRL | NDVI | CTRL | NDVI |
| AgiaMarina_Xyliatou | -0.188 | -0.185 | 0.226 | 0.224 | -0.005 | 0.001 | 1.825 | 1.780 | -1.828 | -1.767 |
| Cairo_EMA_2 | -0.355 | -0.344 | 0.406 | 0.399 | **-0.053** | 0.018 | 1.689 | 1.646 | -1.687 | -1.591 |
| Eilat | -0.138 | 0.006 | 0.186 | 0.165 | **0.110** | **0.312** | 1.183 | 0.610 | -1.166 | 0.034 |
| El_Farafra | -0.186 | -0.190 | 0.259 | 0.263 | **0.170** | **0.138** | 1.155 | 1.248 | -1.218 | -1.257 |
| KAUST_Campus | -0.245 | 0.152 | 0.322 | 0.376 | **0.412** | **0.386** | 0.966 | 0.609 | -1.001 | 0.342 |
| Kuwait_University | -0.097 | 0.007 | 0.275 | 0.278 | **0.152** | **0.266** | 0.588 | 0.537 | -0.290 | 0.018 |
| Mezaira | -0.130 | 0.161 | 0.228 | 0.347 | **0.353** | **0.445** | 0.528 | 0.475 | -0.382 | 0.332 |
| SEDE_BOKER | -0.151 | -0.125 | 0.198 | 0.201 | 0.030 | 0.034 | 1.202 | 1.209 | -1.228 | -0.921 |
| Weizmann_Institute | -0.207 | -0.180 | 0.264 | 0.255 | **-0.088** | **-0.100** | 1.494 | 1.323 | -1.521 | -1.197 |





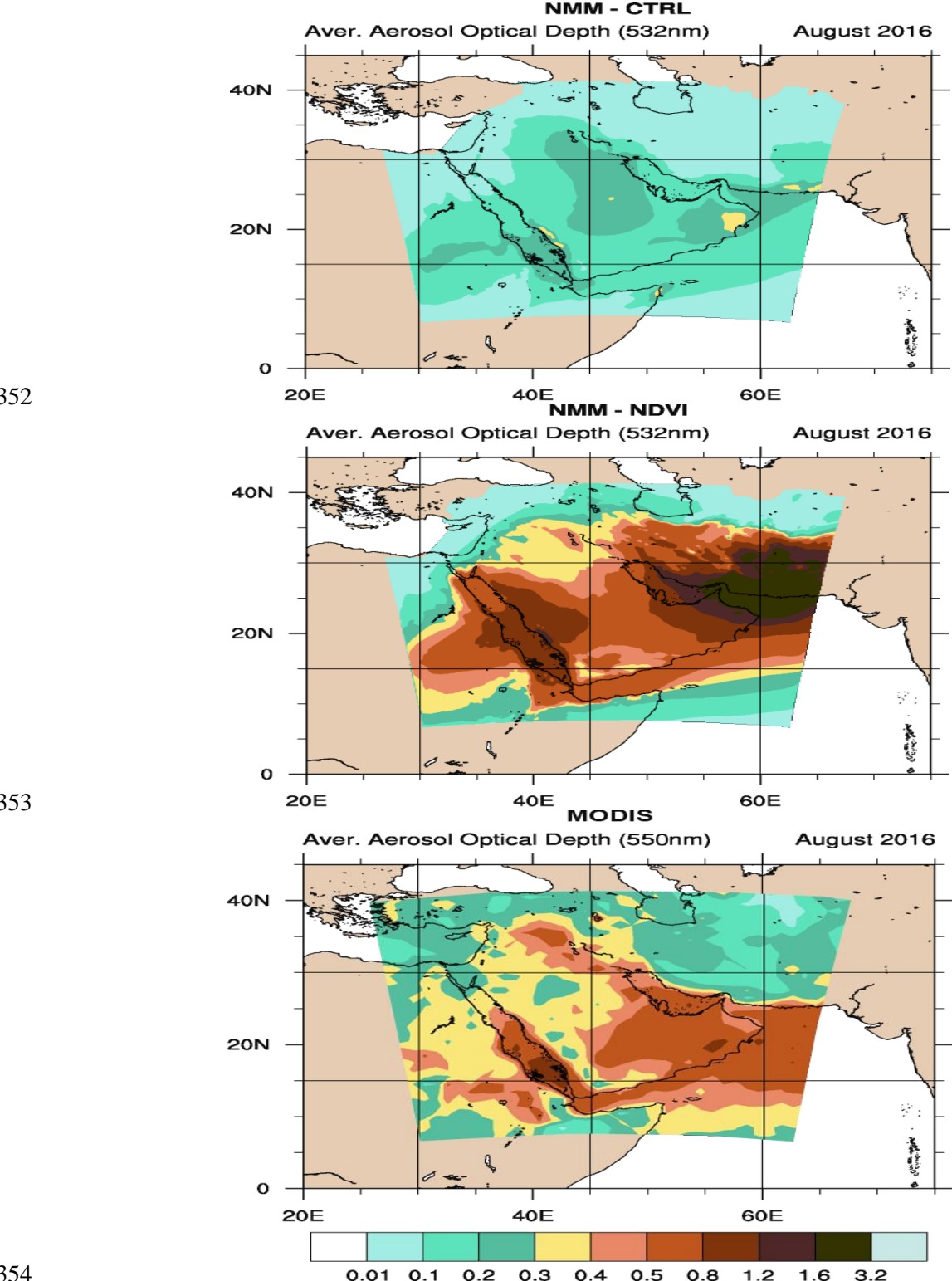

Figure 4. Monthly average simulated AOD during August 2016 from DREAM-CTRL run (a),
DREAM-NDVI run (b) and (c) MODIS. The dashed trapezoid in (c) denotes the location of the
modeling domain.

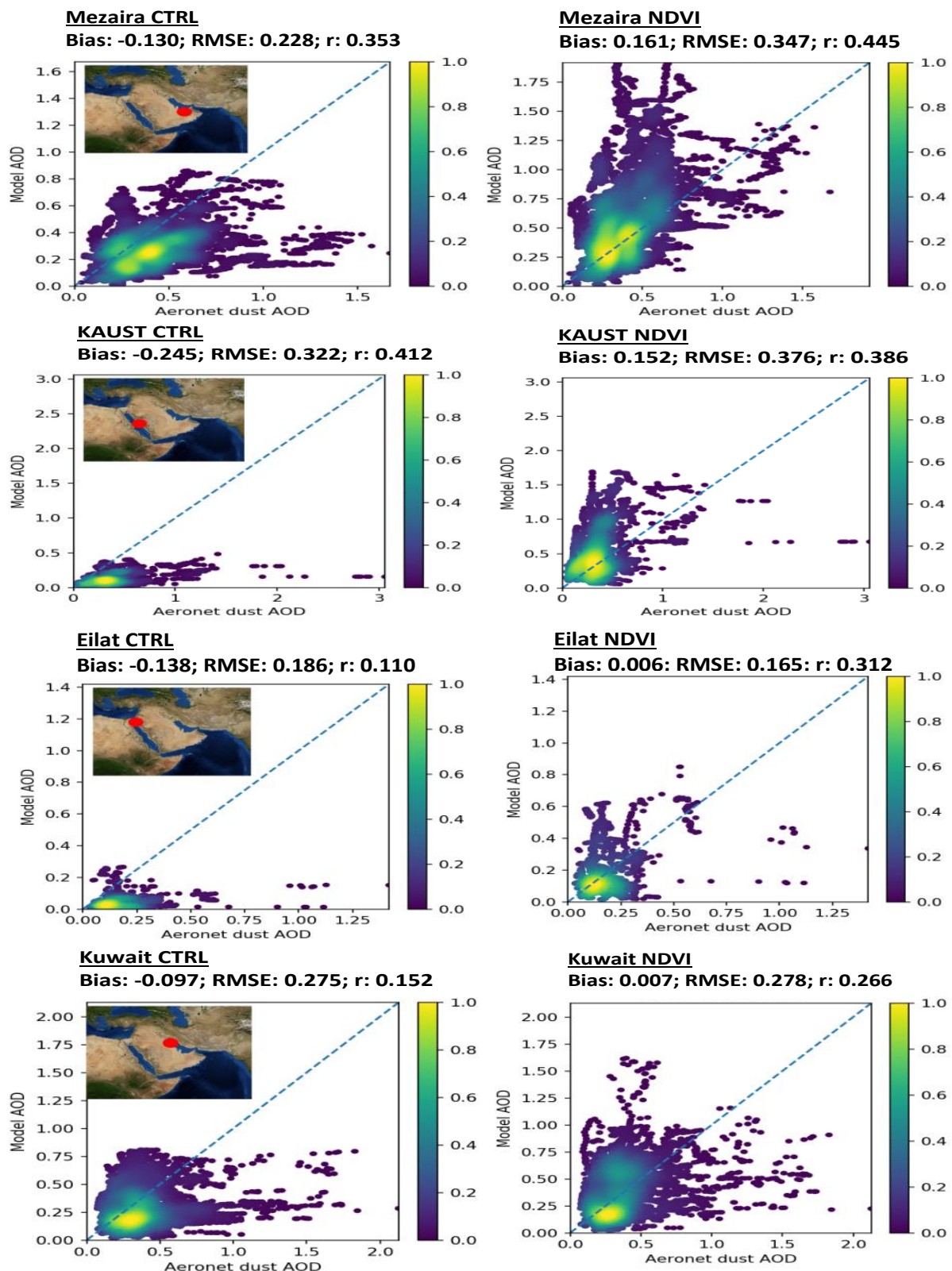


Figure 5. Density scatter plots of modeled and AERONET dust AOD at the stations of Mezaira,
Kaust, Eilat and Kuwait for 2016.

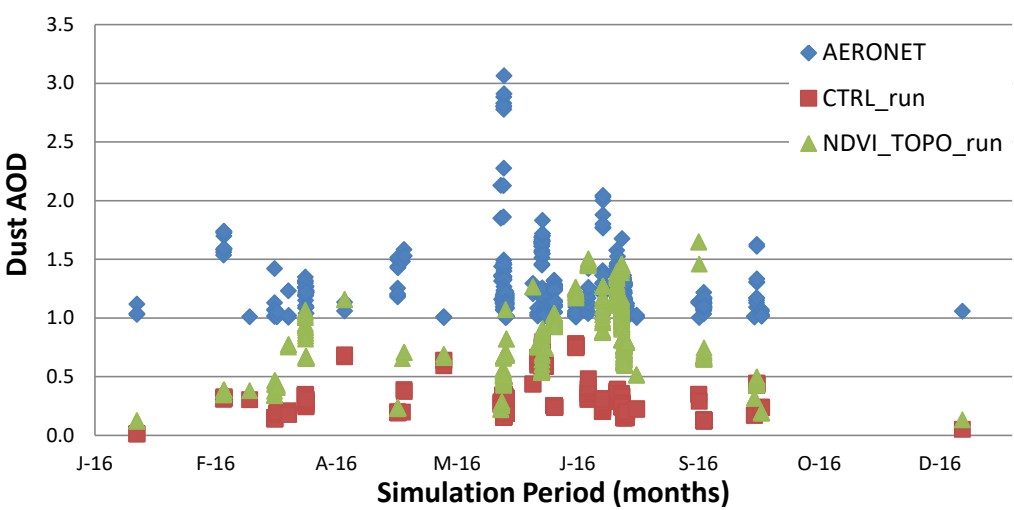

Figure 6. Timeseries of measured and modeled dust AOD for the cases of AERONET AOD>1

**Code and Data availability**

All code and data used in this study are available upon request.

**Author Contribution**

SS: Conceptualization, Formal analysis, Investigation, Methodology, Project administration, Resources, Software, Validation, Visualization, Writing - original draft, Writing – review & editing;

AA: Conceptualization, Funding acquisition, Project administration, Supervision, Writing – review & editing;

CS: Software, Data curation, Visualization, Writing – review & editing;

IB: Conceptualization, Formal analysis, Software, Writing – review & editing;

SN: Methodology, Supervision, Writing – review & editing;

**Acknowledgements**

This work was funded by a grant from the National Space Science and Technology Center of the United Arab Emirates University under grant number NSS Center 7 -2017.The authors acknowledge also support from BEYOND Centre of Excellence (FP7-REGPOT-2012-2013-1, grant agreement no. 316210) for providing financial support and computing resources.

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
