# Peer review of "Development of a dynamic dust-source map for NMME-DREAM v1.0 model based on MODIS NDVI over the Arabian Peninsula"

_Geoscientific Model Development, 2018_

## Referee Comment (RC1) · Parajuli (Referee) · 4 Dec 2018

The paper proposes a dynamic dust source map based on NDVI values and use it in a dust model. The "dynamics" of dust sources is an important aspect of dust emission, which is currently not represented in many dust models. In this regard, this paper attempts to address an important concern of the dust modeling community. The paper is written concisely with great focus, which I greatly appreciate. However, some important relevant descriptions required are missing in the manuscript. In addition, it is not clear if the results dictate enough to justify the use of dynamic dust source map. My specific comments are given below:

[Figure]

Line 52-59. It is mentioned that dust sources are represented by global datasets but it is not described 'how' exactly they are represented. Please describe how it is done, at least in DREAM. Please explain how exactly dust emission is affected when we use the new dust source map in the DREAM model. Please show and explain the detail of the particular equation that is affected. I believe the main change is the fraction of a grid point covered by desert surface, as in equation 3 of Nicovic et al., (2001).

60-73: You describe other works that used dynamic dust source map but it is not clear how your 'proposed' work is similar/different to these previous works. Please make it clearer.

78-79, in the 1) control run, do you use Ginoux et al. 2001 source map or Olson dataset? Please clarify the link between these two datasets.

91-99. this description is not so relevant to this study. It is not necessary to talk about partial differential equations or turbulent parameters here. This study is more about the dust source characterization so there should be more background or description from 'dust-source' point of view. A few sentences about the overall model description is sufficient. Rather, a brief description and comparison of different dust emission models currently in use, e.g., GOCART (WRF), DEAD (CESM), MACC etc., would be helpful for the readers. Also the literature on dust source map should be extended to cover the most recent developments in this topic. Some relevant starting references are given below: Parajuli, S. P and C. Zender (2017), Connecting geomorphology to dust emission through high-resolution mapping of global land cover and sediment supply, Aeol. Res., 27, pp. 47-65, doi:10.1016/j.aeolia.2017.06.002. Ginoux, P., J. M. Prospero, T. E. Gill, N. C. Hsu, and M. Zhao (2012), Global-scale attribution of anthropogenic and natural dust sources and their emission rates based on MODIS deep blue aerosol products, Rev. Geophys., 50(3), doi:10.1029/2012RG000388.

Line 103: Make it clear that August 2016 is test run but simulations are conducted for the whole year.

[Figure]

107-108: is it updated monthly or every 16 days? In many models, it is generally updated monthly. Please clarify. Also please describe the 'numerical procedure' that you mention in more detail explaining how the NDVI data is used to define the dust sources.

112/113: Is this ratio a modification from Nicovic et al., 2001, equation 3? Please clarify.

120: how are those mountainous areas removed? Western Saudi region has many dust sources with intermountain deposits; make sure that you do not overlook these sources. See below for those dust sources. Anisimov et al. (2017), Quantifying local-scale dust emission from the Arabian Red Sea coastal plain, Atmospheric Chemistry and Physics, 2017;17(2):993-1015 DOI 10.5194/acp-17-993-2017.

Line 146: How is effective strength of dust source defined? It is not described clearly.

155-165: This paragraph belongs to the 'methods' section, not the results section. So it should appear early in the manuscript.

Line 156/157. Please describe how exactly the static Ginoux et al., 2001 dust source map was used in the model previously? Is the dust emission equation 'tuned' to achieve a target AOD as commonly used in many dust models? The comparison would be better if the control and NDVI run both were tuned to achieve some observed AOD value. Was the model tuned in some way to get a desired AOD? Figure 2b. Is this map showing the values from A_gridbox that you defined earlier? Please clarify what exactly are the plotted values. Figure 3 titles: May be DREAM-CTRL and DREAM-NDVI are better titles? Figure 4. Please describe Figure 4 in the text properly. Figure 5. Please describe Figure 5 in the text properly. Is it NDVI_run or NDVI_topo_run in the legend? I think you use topographic source function (Ginoux et al., 2001) in the control run so it is confusing. You may not need to set the color bar maximum value to 6.4, which is very high. A lower value of 1-2 would be sufficient.

[Figure]

About the evaluation metrics used in the paper: This paper is about the benefit of representing 'dynamics' of dust sources. So the time-correlations should increase if the new changes are beneficial. Improvement in bias does not confirm that it is because of the better representation of the 'dynamical' aspect of dust sources. The simulated values of AOD and their range also affect the bias, which are sensitive to the process of model tuning. That is why I mentioned about tuning previously. In addition, the RMSE is reduced only in one case and it is increased in other three cases? What does this tell? We should perhaps think of better control and model experiments so that the comparison of the two is fairer and the difference will show the expected outcome.

Table 1. What are fractional gross errors and mean fractional bias? Why are they relevant here? I think it is not necessary to show these values.

Some minor typing errors: Line 6. Normalized difference . . . Line 9. One year Line 31. Precipitation process Line 68. ..be even .. Line 248/249: rewrite the first sentence, the main purpose . . . . . .

---

## Referee Comment (RC2) · Terradellas (Referee) · 18 Dec 2018

Manuscript: Development of a dynamic dust-source map for NMME-DREAM v1.0 model based on MODIS NDVI over the Arabian Peninsula

Authors: Solomos Stavros, Abdelgadir Abuelgasim, Christos Spyrou, Ioannis Binietoglou, Slobodan Nickovic

Reviewer: Enric Terradellas

Overall recommendation: Accept subject to minor revision

Rationale: The dust source map is one of the key aspects of the parameterization of

the dust processes into NWP models. One of the biggest problems in the definition of sources is that soil conditions for dust emission change over time, cyclically through-out the year and with much less predictable inter-annual variations. Therefore, the introduction of dynamic dust source maps, based on satellite remote sensing products, seems a necessary step forward. The authors describe the implementation of a map of this type and show that with it the model considerably improves its performance.

Comments for authors:

• Introduction. You should mention here previous attempts to scale the dust emissions by satellite NDVI that you mention in Section 3 (Summary and Discussion). You should emphasize the difference of your approach (if there is any). • page 1, line 7: The expression 'The new modeling system' seems excessive, when you only change the dust source map. • page 1, line 7: ". . . is tested for the analysis of dust particles dispersion...". I assume that the model simulates emission and deposition, not only dispersion. Moreover, the new map influences dust emission. So, why do you write you analyse dispersion? • page 1, line 13: "The modeled AOD bias is improved from -0.140 to 0.083". It is not necessarily an improvement, since you compare dust AOD with total AOD. A positive bias is always bad news, whereas a bias of -0.14 may be acceptable. Going from -0.933 to -0.424 that does seem an improvement. • Page 1, line 27: Some spaces between words are missed: 'studies.Dust' in page 1, line 27; '(CCN)and' or 'precipitationprocesses' in page 1 line 31 and so on. Please, check it. • Page 2 line 54: define 'SSib' • Page 2 line 69. I would not say that the Arabian Peninsula is a good example of heterogeneous region. • Section 1.1. Although there are references on it, I would include a short paragraph describing the main aspects of the dust model (emission, deposition schemes, whether or not there is radiative feedback, ...) • Page 3 line 93: I would suggest 'nonlinear partial differ-ential equation' instead of 'partial differential nonlinear equation' • Page 3 line 111: ' TheNDVI dataset is at finer resolution than the model grid'. Which is the resolution? • Page 3 line 117: 'we have applied a limit to dust efficiency over high mountain'. I

think it would be worth explaining it a little. A limit on a threshold altitude? • Page 3 line 124 & page 4 line 152: In other parts of the text, you mention that the NDVI product is a 16-day average. Here, you present it as a monthly product. Please, clarify. • Page 4 line 137. Please, check the ratio's denominator in the equation below. • Page 4 line 138. 'top of the atmosphere reflectance'. Shouldn't be surface reflectance (measured from the top of the atmosphere)? • Page 4 line 144: 'The high resolution masks was used... '. I don't understand what these masks are. I suppose you use the equation of line 113. In any case, '. . . masks were used...' • Page 5 line 175: 'Arabian Gulf'. I don't want to participate in a naming dispute, but in most international treaties, documents and maps, this body of water is known by the name of Persian Gulf. • Page 6 line 184: I suppose you should comment on the deficiencies of the monthly average of MODIS AOD. I mean that the daily datasets do not cover, far from it, the entire territory due to the presence of clouds, excessive zenithal angle, etc. • Page 6 line 195: 'miss-classification of Iran and Pakistan grid points'. The overestimation along the Iran-Pakistan coastline is very striking. Can you guess a possible cause of this miss-classification here? • Page 6 line 200: 'measurements of AOD?. Please, replace measurements with retrievals. • Section 2.2: In the first paragraph you compare monthly averages of MODIS AOD with monthly averages of simulations. You should explain what you compare in the second paragraph: montly averages, timeseries with individual retrievals, daily averages, ... • Page 7 line 218. Please, re-phrase the sentence starting with 'The bias reverse is evident ...' • Page 7 line 236. the sentence starting with 'These may include bog, marsh, ...' should be revised. Probably you refer to dried or dessicated bog, marshes, ... • Different formats are used for citations. Please, check it.
* * *

---

## Referee Comment (RC3) · Anonymous Referee #3 · 23 Dec 2018

Review for "Development of a dynamic dust-source map for NMME-DREAM v1.0 model based on MODIS 1 NDVI over the Arabian Peninsula" by Stavros et al.

The authors developed a dynamic dust source map based on MODIS Normalized Digital Vegetation Index (NDVI) for the dust emission scheme in the NMME-DREAM v1.0 model over the Arabian Peninsula. Two groups of simulations are conducted for 2016, one with the dynamic source map (NDVI_run) and the other with the default static source map (CTRL_run). It was found that when using the dynamic dust source the simulated AOD biases are reduced for dust episodes (i.e., when AOD>1) in comparison with the simulation using default setting. This paper explored the influence of the

seasonal variation of vegetation coverage on dust emission scheme, which is a very interesting and important topic, and tested their methods over one major dust source regions, Arabian Peninsula. However, the overall presentation needs some improvement, some details need further clarification, and I also have some concerns about the methodology. My comments are listed below.

Major comments:

1. In the introduction part when reviewing previous studies of dust source map, I think it is important to briefly introduce Ginoux et al. (2001), who determined dust source mainly based on topographic depressions. As mentioned in the later part of the paper, this is also the default setting used in the NMME-DREAM model. It is also informative to explain what's new in the method used here compared with previous studies that also used NDVI to develop dust source map in the introduction section. And similarly, in the result section, it is better to discuss current results within the context of previous work that also compared static dust source map with NDVI based source map in this region.

2. Some details regarding the methodology need further clarification, for instance:

a) I think it will be informative to provide the equation of dust emission scheme in the NMME-DREAM model in section 1.1, so readers can see the role of the dust source function.

b) How do you define "# of dust points" in your equation of "Agrid_box"?

c) How do you define dust efficiency in line 120 and "fractional gross error" and "mean fractional bias" in Table 2?

d) Section 1.2 has a lot of redundant lines, e.g., lines 131-133 are the same as lines 140-142, while lines 128, 135 and 143 repeated the same information.

e) According to lines 183-184, it is not clear if the simulated AOD is purely dust AOD, or it also includes the optical depth contributed by other aerosol particles?

[Figure]

f) Line 162 seems indicating that the model settings are different for the CTRL_run and NDVI_run? Is this true?

g) It is also important to briefly introduce the datasets used for model validation in section 1, e.g., the MODIS AOD, AERONET AOD. What are the spatial and temporal resolutions?

3. Two major differences between Fig. 2a and b are the discrepancies of dust source strength over western Saudi Arabia and over Iran and western Pakistan. I think the authors should discuss these differences in the end of section 1.1 and also correspondingly in the result section. It seems to me that the NDVI source map overestimates the dust source strength over western Saudi Arabia and consequently led to too much AOD in this region in Fig. 3b.

4. Section 2.1 discusses dust transport by atmospheric circulation in August 2006. First of all, it is not clear to me whether information presented here is based on model simulation or reanalysis or observational data. Please clarify. On the other hand, those weekly variations of surface winds and dust transport may not necessarily be revealed in the monthly AOD map in Fig. 3. I'd suggest either adding figures of weekly variations of wind and AOD in this section or adding monthly surface wind vectors in Fig. 3 to discuss how winds affect AOD pattern.

5. As you mentioned in lines 233-236, NDVI mask dose not have much seasonal variations in permanent deserts, but may be important in those semi-arid regions, as also pointed by Kim et al. (2013). I wonder if you can also plot 12-month NDVI map in this region for 2016 to demonstrate the influences of NDVI seasonal cycle and then you can discuss the seasonal variations of AOD in Fig. 5 along with NDVI seasonal cycle.

6. The overall magnitude of AOD in the control run is quite low but does seem to have relatively higher values over the eastern Arabian Peninsula, which is consistent with the pattern of MODIS AOD. I wonder if you tried to tune the model in the CTRL simulation

to increase the overall magnitude of dust emission and then compare the pattern and seasonal cycle of AOD with the NDVI_run.

7. Kim et al. (2013) combined both the topographic depression-based dust source and NDVI seasonal masking for dust source map. I wonder if you can combine the dynamic source developed here with the default Ginoux et al. (2001) static source, and see if the model performance is further improved. I think those high AOD over western Saudi Arabia probably will be largely reduced.

8. Here only modeled AOD in the two simulations are compared. I wonder if you also see any improvement in other aspects of dust life cycle such as surface dust concentration, vertical distribution, and deposition.

Minor points:

1. Line 28, add "e.g.," before "Torge et al., 2011"

2. Line 31, add space between "precipitation" and "processes". Please fix all similar occurrences.

3. Line 45, I don't think there is any "feedbacks" on "human health", please consider reorganizing the sentence.

4. Line 63, please add brackets for "2013", and fix all similar occurrences.

5. Line 79, I think the original dust source function developed by Ginoux et al. (2001) did not use "Olson World Ecosystems dataset". Can you explain a bit more here?

6. Line 109, you may want to add a line or two to explain why NDVI of 0.1 is selected instead of 0.15 as used by Kim et al. (2013).

7. Lines 200-201, not clear. Did you use Ångström exponent to mask AOD? In that case, the masked AOD may contain large particles such as dust and sea salt.

8. Fig. 3, it is better to mask out AOD outside the model domain in Fig. 3c for an easy

comparison among the three plots.

9. Please clarify in Table 1 caption that this is for annual mean. And for correlation, do you use monthly data? Can you also mark whether the correlation coefficients are statistically significant?

10. Table 2, are monthly or daily data used for correlation? Please add significance test as well.

11. Fig. 5, are the time series calculated from single AERONET site (which one?) or averaged over four stations on the Arabian Peninsula?

References:

Ginoux, P., Chin, M., Tegen, I., Prospero, J. M., Holben, B., Dubovik, O., & Lin, S.-J., Sources and distributions of dust aerosols simulated with the GOCART model. J. Geophys. Res., 106(D17), 20, 255–20, 273, https://doi.org/10.1029/2000JD000053, 2001

Kim, D., M. Chin, H. Bian, Q. Tan, M. E. Brown, T. Zheng, R. You, T. Diehl, P. Ginoux, and T. Kucsera, The effect of the dynamic surface bareness on dust source function, emission, and distribution, J. Geophys. Res. Atmos., 118, 871–886, doi: 10.1029/2012JD017907, 2013.

---

## Author Comment (AC1) · 24 Jan 2019

The paper proposes a dynamic dust source map based on NDVI values and use it in adust model. The "dynamics" of dust sources is an important aspect of dust emission, which is currently not represented in many dust models. In this regard, this paper attemptsto address an important concern of the dust modeling community. The paper is written concisely with great focus, which I greatly appreciate. However, some important required are missing in the manuscript. In addition, it is not clearif the results dictate enough to justify the use of dynamic dust source map. My specific comments are given below.

[REPLY] We thank the reviewer for the constructive comments. The general purpose of our work is to describe the development of this alternative method to our usual static representation of dust sources in DREAM. It is true that this first application is encouraging but cannot justify the replacement of current model configurations over different source regions. More tests for other areas and periods will be required with the new model setup before we conclude to an optimal dust source description which in the end might be a combination of both static and dynamic maps.

Line 52-59. It is mentioned that dust sources are represented by global datasets but it is not described 'how' exactly they are represented. Please describe how it is done, at least in DREAM. Please explain how exactly dust emission is affected when we usethe new dust source map in the DREAM model. Please show and explain the detail of the particular equation that is affected. I believe the main change is the fraction of agrid point covered by desert surface, as in equation 3 of Nicovic et al., (2001).

[REPLY] We have updated the model description section to include more information: "The original classification of dust sources in DREAM is based on Ginoux et al., (2001) that takes into account the preferential sources related to topographic depressions and paleolake sediments. The global mapping of dust sources in Ginoux et al.,(2001) is determined from the comparison between the elevation of surface grid points at 1°×1° resolution with the surrounding hydrological basins and with the 1°×1° AVHRR (Advanced Very High Resolution Radiometer) vegetation map (DeFries and Townshend, 1994)."In the new version the ratio of arid and semi-arid vegetation points to the total vegetation points inside a model grid-box is used to calculate the corresponding dust productivity of each particular cell.

**60-73: You describe other works that used dynamic dust source map but it is not clear how your 'proposed' work is similar/different to these previous works. Please make itclearer.**

[REPLY]Our work is similar to Kim et al., 2013, Vukovic et al., 2014 and Solomos et al., 2017 in terms of the general objectives and methodology. We have added a revised section: "The main

differences in our approach compared to the previous studies referenced above, is that we use a very high resolution NDVI product (500×500 m) in a regional modeling domain (e.g. Kim et al., 2013 used an 8×8 Km NDVI dataset extrapolated to 1°x1° global modeling domain) and our study is not limited to specific test cases (like for example Vukovic et al., 2014 and Solomos et al., 2017), but covers an extended time period, as presented below."

**78-79, in the 1) control run, do you use Ginoux et al. 2001 source map or Olson dataset? Please clarify the link between these two datasets.**

[REPLY] Thank you for pointing this out. We use Ginoux et al., 2001. We have revised this sentence as: "Control run, where the dust source definition is based on the Ginoux et al., (2001) dataset"

91-99. this description is not so relevant to this study. It is not necessary to talk about partial differential equations or turbulent parameters here. This study is more about the dust source characterization so there should be more background or descriptionfrom 'dust-source' point of view. A few sentences about the overall model descriptionis sufficient. Rather, a brief description and comparison of different dust emissionmodels currently in use, e.g., GOCART (WRF), DEAD (CESM), MACC etc., would be helpful for the readers. Also the literature on dust source map should be extended to cover the most recent developments in this topic. Some relevant starting references are given below:

Parajuli, S. P and C. Zender (2017), Connecting geomorphology to dust emission through highresolution mapping of global land cover and sediment supply, Aeol. Res., 27, pp. 47-65, doi:10.1016/j.aeolia.2017.06.002.

Ginoux, P., J.M. Prospero, T. E. Gill, N. C. Hsu, and M. Zhao (2012), Global-scale attribution of anthropogenic and natural dust sources and their emission rates based on MODIS deep blue aerosol products, Rev. Geophys., 50(3), doi:10.1029/2012RG000388.

[REPLY] We have increased the information on the surface dust source map considerations. The revised section is now:

"The original classification of dust sources in DREAM is based on Ginoux et al., (2001) that takes into account the preferential sources related to topographic depressions and paleolake sediments. The global mapping of dust sources in Ginoux et al.,(2001) is determined from the comparison between the elevation of surface grid points at 1°×1° resolution with the surrounding hydrological basins and with the 1°×1° AVHRR (Advanced Very High Resolution Radiometer) vegetation map (DeFries and Townshend, 1994). Recent studies indicated the contribution of both natural and anthropogenic dust sources to the overall dust emissions detected in MODIS Deep Blue product (Ginoux et al., 2012) and also the relevance of local geomorphological conditions and sediment supply (Parajuli and Zender, 2017) on the global dust emissions. All these advances in dust emissions are based on static map considerations."

**Line 103: Make it clear that August 2016 is test run but simulations are conducted for the whole year.**

[REPLY] This sentence is revised as follows: "August 2016 has been selected as a test period for the model development due to the significant dust activity and variability in wind properties

during this month. One-year runs for the entire 2016 have been conducted to evaluate the performance of the static and dynamic database emission maps".

**107-108: is it updated monthly or every 16 days? In many models, it is generally updated monthly. Please clarify. Also please describe the 'numerical procedure' that you mention in more detail explaining how the NDVI data is used to define the dustsources.**

[REPLY] Yes we used the 500×500 m 16-day averaged NDVI from MODIS. We assume that regions with NDVI values from 0 to 0.1 correspond to bare soil and therefore can be efficient sources. The NDVI dataset is at finer resolution than the model grid and in order to find the potential for dust production in each model grid box, we calculate the ratio: number\_of\_dust\_points / total points. The scaling of satellite data over model grid points allows the use of the same algorithm for different model configurations.

**112/113: Is this ratio a modification from Nicovic et al., 2001, equation 3? Please clarify.**

[REPLY] Yes, it is formally of the same kind but the input parameters are different. Instead of desert, semidesert, arid and semiarid vegetation points we use NVDI<0.1 points.

**120: how are those mountainous areas removed? Western Saudi region has many dust sources with intermountain deposits; make sure that you do not overlook these sources. See below for those dust sources.**

Anisimov et al. (2017), Quantifying local scale dust emission from the Arabian Red Sea coastal plain, Atmospheric Chemistry and Physics, 2017;17(2):993-1015 DOI 10.5194/acp-17-993-2017.

[REPLY] The following paragraph has been added in the revised version: "In order to exclude such unrealistic emissions from non-soil bare areas or snow-covered areas we have applied a limit of zero dust production above 2500 m over the entire domain. This simple approach has been selected in order to keep our straightforward NDVI mapping independent of vegetation and soil information. The threshold value of 2500 m does not suppress the emissions from lowlands and hillsides (e.g. the coastal areas of Hejaz Mountains in Red Sea that have been identified as hot dust spots by Anisimov et al., 2017)."

**Line 146: How is effective strength of dust source defined? It is not described clearly.**

[REPLY] "The high-resolution mask was used to calculate the percentage of barren land in each  $0.1^{\circ}x0.1^{\circ}$  model grid cells and this percentage was used to define Agrid box in Eq.1"

**155-165: This paragraph belongs to the 'methods' section, not the results section. Soit should appear early in the manuscript.**

[REPLY] Done. We moved this paragraph to Section 1.1 (Model Description).

Line 156/157. Please describe how exactly the static Ginoux et al., 2001 dust sourcemap was used in the model previously? Is the dust emission equation 'tuned' to achieve a target AOD as commonly used in many dust models? The comparison would bebetter if the control and NDVI run both were tuned to achieve some observed AODvalue. Was the model tuned in some way to get a desired AOD?

[REPLY] The control run (CTRL) is our standard configuration and it is tuned to reproduce as much as possible the AOD over both African and Asia sources. The NDVI run is performed with exactly the same configuration other than the definition of dust sources (Eq.1).

**Figure 2b. Is this map showing the values from A\_gridbox that you defined earlier? Please clarify what exactlyare the plotted values.**

[REPLY] Yes. This in now clarified in the caption.

**Figure 3 titles: May be DREAM-CTRL and DREAM-NDVI are better titles?**

[REPLY] Yes indeed, thank you. We have changed these titles throughout the revised text.

**Figure 4. Please describe Figure 4 in the text properly.**

[REPLY] In the revised manuscript we have changed lines 199-200 to: "The modeled dust optical depth is compared with the regional AERONET ground-based photometric measurements of AOD (Figure 4)...."

**Figure 5. Please describe Figure 5 in the text properly. Is it NDVI\_run or NDVI\_topo\_run in the legend? I think you use topographic source function (Ginoux et al., 2001) in the control run so it is confusing.**

[REPLY] In the revised manuscript we use DREAM\_CTRL and DREAM\_NDVI instead.

**You may not need to set the color bar maximum value to 6.4, which is very high. A lower value of 1-2 would be sufficient.**

[REPLY] This probably refers to Figure 3. This color bar is now revised in this plot. A maximum value of 3.2 is selected, as this is the maximum value from the simulations output.

About the evaluation metrics used in the paper: This paper is about the benefit of representing 'dynamics' of dust sources. So the time-correlations should increase if the new changes are beneficial. Improvement in bias does not confirm that it is because of the better representation of the 'dynamical' aspect of dust sources. The simulated values of AOD and their range also affect the bias, which are sensitive to the process of model tuning. That is why I mentioned about tuning previously. In addition, the RMSE is reduced only in one case and it is increased in other three cases? What does this tell?

[REPLY]An overall tuning factor is often applied homogeneously over modeling domains. This can only result in a linear emission increase which may benefit the model results in one area but deteriorate the statistics in other areas. This is not the case here since the only change in the model (Eq.1) is the replacement of dust sources (Ginoux et al., 2001) with the NDVI dust points. The RMSE is increased for the DREAM\_NDVI run due to the increase in maximum modeled AODs. For the severe dust episodes (AOD>1) the RMSE is improved.

We should perhaps think of better control and model experiments so that the comparison of the two is fairer and the difference will show the expected outcome.

[REPLY] In our opinion, the method presented in our work clearly shows the potential of satellite retrievals as an alternative method for the mapping of dust sources. In general, we believe that dust emissions should be described in atmospheric models based only on physical considerations without the need for empirical tuning factors. In this direction a combination of up to date and detailed land cover mapping with synchronous remote sensing information (e.g. NDVI from various sensors) could lead to better results in future work.

**Table 1. What are fractional gross errors and mean fractional bias? Why are they relevant here? I think it is not necessary to show these values.**

[REPLY] These are common statistical metrics used for example by the WMO SDS-WAS system and are included here for consistency with the operational evaluation of dust models. More description has been added in the revised text.

Some minor typing errors: Line 6. Normalized difference Line 9. One year Line 31.Precipitation process Line 68. ..be even .. Line 248/249: rewrite the first sentence, themain purpose

[REPLY] Done, thank you.

---

## Author Comment (AC2) · 24 Jan 2019

**Reply to the reviewer 2 comments**

**Terradellas (Referee)**

eterradellasj@aemet.es

**Overall recommendation: Accept subject to minor revision**

**Rationale: The dust source map is one of the key aspects of the parameterization of the dust processes into NWP models. One of the biggest problems in the definition of sources is that soil conditions for dust emission change over time, cyclically through- out the year and with much less predictable inter-annual variations. Therefore, the introduction of dynamic dust source maps, based on satellite remote sensing products, seems a necessary step forward. The authors describe the implementation of a map of this type and show that with it the model considerably improves its performance.**

[REPLY] We would like to thank the reviewer for his comments and suggestions. The replies to the specific comments follow:

**Comments for authors:**

**Introduction. You should mention here previous attempts to scale the dust emissions by satellite NDVI that you mention in Section 3 (Summary and Discussion). You should emphasize the difference of your approach (if there is any).**

[REPLY] We have added extended relevant sections in the revised version:

Introduction: "The main differences in our approach compared to the previous studies referenced above, is that we use a very high resolution NDVI product (500×500 m) in a regional modeling domain (e.g. Kim et al., 2013 used an 8×8 Km NDVI dataset extrapolated to 1°x1° global modeling domain) and our study is not limited to specific test cases (like for example Vukovic et al., 2014 and Solomos et al., 2017), but covers an extended time period, as presented below."

Methodology: "The global mapping of dust sources in Ginoux et al.,(2001) is determined from the comparison between the elevation of surface grid points at 1°×1° resolution with the surrounding hydrological basins and with the 1°×1° AVHRR (Advanced Very High Resolution Radiometer) vegetation map (DeFries and Townshend, 1994)."

Summary and Discussion: "These findings support the previous results by Kim et al., 2013 who also showed an increase in dust emissions and a more realistic comparison with satellite observations in Saudi Arabia by the introduction of an NDVI based dynamic source mapping for GOCART model."

**Page 1, line 7: The expression 'The new modeling system' seems excessive, when you only change the dust source map.**

[REPLY] The sentence has been changed to read: "The updated modeling system."

**Page 1, line 7: ". . . is tested for the analysis of dust particles dispersion...". I assume that the model simulates emission and deposition, not only dispersion. Moreover, the new map influences dust emission. So, why do you write you analyse dispersion?**

[REPLY] The model simulated both emission and deposition and our development actually changes only the emission of dust. The sentence has been changed to read: ". . . is tested for dust emission capabilities..." in order for that to be clear.

**page 1, line 13: "The modeled AOD bias is improved from -0.140 to 0.083". It is not necessarily an improvement, since you compare dust AOD with total AOD. A positive bias is always bad news, whereas a bias of -0.14 may be acceptable. Going from -0.933 to -0.424 that does seem an improvement.**

[REPLY] The reviewer raises an interesting issue. Seeing as absolute numbers a bias of 0.083 is better than -0.140, which is what we write in the text. However, as the reviewer states, an overestimation is not something we want when evaluating dust production and transport. However in more severe cases (AOD>1) the model performs better. We believe that this issue could be resolved by increasing the resolution of the model domain, thus giving a more detailed representation of the dust source areas, something that our methodology allows us to do, since the resolution of NDVI product is already at 500×500 m.

**Page 1, line 27: Some spaces between words are missed: 'studies. Dust' in page 1, line 27; '(CCN) and' or 'precipitation processes' in page 1 line 31 and so on. Please, check it.**

[REPLY] Corrected. Also the whole manuscript is revised and other instances have been corrected. Also some double spaces have also been corrected.

**Page 2 line 54: define 'SSib'**

[REPLY] SSib stands for simplified simple biosphere. This has been added to the text along with its reference: Xue et al. 1991

**Page 2 line 69. I would not say that the Arabian Peninsula is a good example of heterogeneous region.**

[REPLY] We wanted to emphasize the use of the Arabian Peninsula as our area of interest, as it is a more heterogeneous region than the Saharan Desert, which could be used as a test area for our methodology. Of course there are more heterogeneous areas that it could potentially be tested, but the Arabian Peninsula is the second biggest desert area of the world and since we wanted a mix of desert size and soil heterogeneity, we deemed it proper for our work. A small comment has been added to the text.

**Section 1.1. Although there are references on it, I would include a short paragraph describing the main aspects of the dust model (emission, deposition schemes, whether or not there is radiative feedback, ...)**

[REPLY] Section 1.1 has been updated and new references have been added to include additional information about the model processes following the reviewer's suggestion.

**Page 3 line 93: I would suggest 'nonlinear partial differential equation' instead of 'partial differential nonlinear equation'**

[REPLY] We agree. It has been changed in the manuscript accordingly.

**Page 3 line 111: 'The NDVI dataset is at finer resolution than the model grid'. Which is the resolution?**

[REPLY] The resolution of the NDVI dataset is 500×500 m. This has been added to the sentence.

**Page 3 line 117: 'we have applied a limit to dust efficiency over high mountain'. I think it would be worth explaining it a little. A limit on a threshold altitude?**

[REPLY] The following paragraph has been added in the revised version: "In order to exclude such unrealistic emissions from non-soil bare areas or snow-covered areas we have applied a limit of zero dust production above 2500 m over the entire domain. This simple approach has been selected in order to keep our straightforward NDVI mapping independent of vegetation and soil information. The threshold value of 2500 m does not suppress the emissions from lowlands and hillsides (e.g. the coastal areas of Hejaz Mountains in Red Sea that have been identified as hot dust spots by Anisimov et al., 2017)."

**Page 3 line 124 & page 4 line 152: In other parts of the text, you mention that the NDVI product is a 16-day average. Here, you present it as a monthly product. Please, clarify.**

[REPLY] Corrected. The product presented here is the 16 day average from 1st to 16th of August 2016. It has been change in the revised text.

**Page 4 line 137. Please, check the ratio's denominator in the equation below.**

[REPLY] Corrected. It now reads $X_{nir} + X_{red}$

**Page 4 line 138. 'top of the atmosphere reflectance'. Shouldn't be surface reflectance (measured from the top of the atmosphere)?**

[REPLY]. This is actually surface reflectance as would be measured at ground level (i.e. corrected for atmospheric effects). This is now stated more clearly in the revised text.

**Page 4 line 144: 'The high resolution masks was used…'. I don't understand what these masks are. I suppose you use the equation of line 113. In any case, '. . . masks were used…'**

[REPLY] Correct. We change this to high resolution dataset in order to be clear.

**Page 5 line 175: 'Arabian Gulf'. I don't want to participate in a naming dispute, but in most international treaties, documents and maps, this body of water is known by the name of Persian Gulf.**

[REPLY] We changed this phrase to: "over the Red Sea and Gulf regions"

**Page 6 line 184: I suppose you should comment on the deficiencies of the monthly average of MODIS AOD. I mean that the daily datasets do not cover, far from it, the entire territory due to the presence of clouds, excessive zenithal angle, etc.**

[Reply] This is correct. We have added a description and reference to better highlight the limitations of the 16-day NDVI product, especially focusing on the compositing difficulties: "The 16-day composite is calculated by ingesting two 8-day composite surface reflectance granules, while the procedure takes into account pixel quality, presence of clouds, and viewing geometry. This procedure can lead to spatial discontinuities, as it is possible that data from different days are used for adjacent pixels, each representing different measurement conditions. If a pixels has with no useful measurements during the 16-day period, historic data are used as fill values (Didan et al., 2015)."

**Page 6 line 195: 'miss-classification of Iran and Pakistan grid points'. The overestimation along the Iran-Pakistan coastline is very striking. Can you guess a possible cause of this miss-classification here?**

[REPLY] Indeed the overestimation is an issue at that area because of the complexity of the terrain in that area, where barren land changes abruptly to desert and vice versa. As stated in the text this is probably "due to a possible miss-classification of Iran and Pakistan grid points as effective dust sources thus favoring unrealistic southeasterly transport towards the Gulf of Oman."

**Page 6 line 200: 'measurements of AOD?. Please, replace measurements with retrievals**.

[REPLY] Corrected according to reviewer's comment

**Section 2.2: In the first paragraph you compare monthly averages of MODIS AOD with monthly averages of simulations. You should explain what you compare in the second paragraph: montly averages, timeseries with individual retrievals, daily averages,**

[REPLY] In the second paragraph of section 2.2 we compare monthly AOD values. A sentence has been added for clarity: "The first step is to examine how our methodology compares against the monthly average AOD in our study area. Therefore the monthly average AOD values produced from our two simulations (NDVI_run and CTRL_run) are compared."

**Page 7 line 218. Please, re-phrase the sentence starting with 'The bias reverse is evident ...'**

[REPLY] Rephrased. The sentence now reads: "Both cases show an improvement in the bias values over the control simulations. When we consider AOD>1 the NDVI_run still underestimates the observed values, but with a lower RMSE (0.586 versus 0.983 of the CTRL_run)."

**Page 7 line 236.the sentence starting with 'These may include bog, marsh, ...' should be revised. Probably you refer to dried or dessicated bog, marshes,**

[REPLY] Revised according to the reviewer's comment.

**Different formats are used for citations. Please, check it.**

[REPLY] The citations have all been reformatted according to the guidelines of the Journal

---

## Author Comment (AC3) · 24 Jan 2019

Review for "Development of a dynamic dust-source map for NMME-DREAM v1.0 model based on MODIS 1 NDVI over the Arabian Peninsula" by Stavros et al.

The authors developed a dynamic dust source map based on MODIS Normalized DigitalVegetation Index (NDVI) for the dust emission scheme in the NMME-DREAM v1.0 model over the Arabian Peninsula. Two groups of simulations are conducted for 2016,one with the dynamic source map (NDVI_run) and the other with the default static source map (CTRL_run). It was found that when using the dynamic dust source thesimulated AOD biases are reduced for dust episodes (i.e., when AOD>1) in comparisonwith the simulation using default setting. This paper explored the influence of the seasonal variation of vegetation coverage on dust emission scheme, which is a veryinteresting and important topic, and tested their methods over one major dust sourceregions, Arabian Peninsula. However, the overall presentation needs some improvement, some details need further clarification, and I also have some concerns about the methodology. My comments are listed below.

[REPLY] We thank the reviewer for the careful revision and useful comments. Indeed the seasonal variation of dust source efficiency is an important topic. In our study we focus to present an alternative method for dust source classification in the well-established DREAM dust model that will allow the system to incorporate the updated satellite NDVI for describing dust emissions. Specific replies to the reviewer's comments follow below.

Major comments:
1. In the introduction part when reviewing previous studies of dust source map, I think it is important to briefly introduce Ginoux et al. (2001), who determined dust source mainly based on topographic depressions. As mentioned in the later part of the paper,this is also the default setting used in the NMME-DREAM model. It is also informativeto explain what's new in the method used here compared with previous studies thatalso used NDVI to develop dust source map in the introduction section. And similarly, in the result section, it is better to discuss current results within the context of previouswork that also compared static dust source map with NDVI based source map in thisregion.

[REPLY] We have extended the corresponding sections in the revised manuscript by adding the following sentences in the introduction, methodology, summary and discussion sections:

Introduction: "The main differences in our approach compared to the previous studies referenced above, is that we use a very high resolution NDVI product (500×500 m) in a regional modeling domain (e.g. Kim et al., 2013 used an 8×8 Km NDVI dataset extrapolated to 1°x1° global modeling domain) and our study is not limited to specific test cases (like for example Vukovic et al., 2014 and  Solomos et al., 2017), but covers an extended time period, as presented below."

Methodology: "The global mapping of dust sources in Ginoux et al.,(2001) is determined from the comparison between the elevation of surface grid points at 1°×1° resolution with the surrounding hydrological basins and with the 1°×1° AVHRR (Advanced Very High Resolution Radiometer)vegetation map (DeFries and Townshend, 1994)."

Summary and Discussion: "These findings support the previous results by Kim et al., 2013 who also showed an increase in dust emissions and a more realistic comparison with satellite observations in Saudi Arabia by the introduction of an NDVI based dynamic source mapping for GOCART model."

**2. Some details regarding the methodology need further clarification, for instance:**
**a) I think it will be informative to provide the equation of dust emission scheme in the NMME-DREAM model in section 1.1, so readers can see the role of the dust sourcefunction.**

[REPLY]The following section has been added in the revised text: "The surface concentration is calculated using equation (11) from Nickovic et al., (2001):

$$C_{sfc} = c_1 \cdot \delta \cdot u_*^2 \left[ 1 - \left( \frac{u_{*t}}{u_*} \right)^2 \right]$$ where $c_1 = 2.4 \cdot 10^{-4} \frac{Kgr}{m^5 \sec^2}$ a tuning constant determined from

model experiments, $u_*$ and $u_{*t}$ the friction velocity and the threshold friction velocity for dust production respectively and $\delta = a \cdot \gamma_k \cdot \beta_k$ , where $\gamma_k$ the ratio between the mass available for uplift and the total mass $\beta_k$ the fractions of clay, silt and sand for each soil class, and $a$ the desert mask (between 0 and 1) calculated from the Ginoux et al., (2001) dataset."

**b) How do you define "# of dust points" in your equation of "Agrid_box"?**
[REPLY] The number of dust points are those that have NDVI values smaller than 0.1. A sentence has been added in the text to clarify that: "Where $\#\_of\_dust\_po\operatorname{int}s$ is the number of points with NDVI values smaller than 0.1."

**c) How do you define dust efficiency in line 120 and "fractional gross error" and "meanfractional bias" in Table 2?**
[REPLY] By "dust efficiency" we refer to how potent is a certain area in producing dust particles by mechanical processes (wind speed).

Fractional gross error calculated for n pairs of model values ($m_i$) and observations ($o_i$) is defined as

$$FGE = 2 \; \overline{\left| \frac{m_i - o_i}{m_i + o_i} \right|}$$

where the bar denotes the mean value (Boylan and Russell, 2006). Similarly, mean fractional bias is defined as

$$MFB = 2 \frac{\overline{m_i - o_i}}{\overline{m_i + o_i}}$$

following(Chang and Hanna, 2004).

We have added a new section that properly defines the quantities used for model evaluation.

Chang, J. C. and Hanna, S. R.: Air quality model performance evaluation, MeteorolAtmosPhys, 87(1–3), 167–196, doi:10.1007/s00703-003-0070-7, 2004.

Boylan, J. W. and Russell, A. G.: PM and light extinction model performance metrics, goals, and criteria for three-dimensional air quality models, Atmos. Environ., 40(26), 4946–4959, doi:10.1016/j.atmosenv.2005.09.087, 2006.

**d) Section 1.2 has a lot of redundant lines, e.g., lines 131-133 are the same as lines140-142, while lines 128, 135 and 143 repeated the same information.**
[REPLY] Thank you for this notice. The redundant lines have been removed at the revised version.

**e) According to lines 183-184, it is not clear if the simulated AOD is purely dust AOD,or it also includes the optical depth contributed by other aerosol particles?**
[REPLY] The simulated AOD is purely dust AOD. The MODIS AOD may include other aerosol types.

**f) Line 162 seems indicating that the model settings are different for the CTRL_run and NDVI_run? Is this true?**
[REPLY] No, the only difference between the two configurations is the definition of dust sources. This line is now removed for clarity.

**g) It is also important to briefly introduce the datasets used for model validation insection 1, e.g., the MODIS AOD, AERONET AOD. What are the spatial and temporal resolutions?**
[REPLY] We have added a new section (1.3 in the revised manuscript) describing the validation datasets.

**3. Two major differences between Fig. 2a and b are the discrepancies of dust sourcestrength over western Saudi Arabia and over Iran and western Pakistan. I think the authorsshould discuss these differences in the end of section 1.1 and also correspondinglyin the result section. It seems to me that the NDVI source map overestimatesthe dust source strength over western Saudi Arabia and consequently led to too muchAOD in this region in Fig. 3b.**
[REPLY] The following lines have been added in the revised text:
"The two dust source patterns present remarkable difference especially over the western Saudi Arabia and over Iran and Pakistan where the NDVI classification results in stronger emissions."
"The DREAM-NDVI AOD is also higher than MODIS AOD over western Saudi Arabia indicating a possible overprediction of dust sources at this area."

**4. Section 2.1 discusses dust transport by atmospheric circulation in August 2006.First of all, it is not clear to me whether information presented here is based on modelsimulation or reanalysis or observational data. Please clarify. On the other hand, thoseweekly variations of surface winds and dust transport may not necessarily be revealedin the monthly AOD map in Fig. 3. I'd suggest either adding figures of weekly variationsof wind and AOD in this section or adding monthly surface wind vectors in Fig. 3 todiscuss how winds affect AOD pattern.**

[REPLY] We have introduced a new plot (Figure 3 in the revised version) that shows the average modeled wind speed and vectors for August 2006 in order to facilitate the corresponding discussion.

**5. As you mentioned in lines 233-236, NDVI mask do not have much seasonalvariations in permanent deserts, but may be important in those semi-arid regions, asalso pointed by Kim et al. (2013). I wonder if you can also plot 12-month NDVI mapin this region for 2016 to demonstrate the influences of NDVI seasonal cycle and thenyou can discuss the seasonal variations of AOD in Fig. 5 along with NDVI seasonalcycle.**

[REPLY] The main purpose of our study is to provide a dynamic modeling tool for dust source definition in NMME-DREAM v1.0 model and demonstrate its capability as an alternative method. Therefore we intend to constrain our work to the description of our proposed methodology. A more in depth analysis of the seasonal dust source variability of at the area would require a longer study period and will be the scope of a forthcoming study.

**6. The overall magnitude of AOD in the control run is quite low but does seem to haverelatively higher values over the eastern Arabian Peninsula, which is consistent with thepattern of MODIS AOD. I wonder if you tried to tune the model in the CTRL simulation to increase the overall magnitude of dust emission and then compare the pattern andseasonal cycle of AOD with the NDVI_run.**

[REPLY] The default configuration is similar to the operational model setup used for example in SDS-WAS (https://sds-was.aemet.es/) and BEYONDhttp://beyond-eocenter.eu/dusthub/, which is tuned towards stations at Africa, Asia and Europe. It is also important to notice that our proposed method is not a simple homogeneous tuning factor but an overall different treatment of dust source definitions.

**7. Kim et al. (2013) combined both the topographic depression-based dust source andNDVI seasonal masking for dust source map. I wonder if you can combine the dynamicsource developed here with the default Ginoux et al. (2001) static source, and see ifthe model performance is further improved. I think those high AOD over western SaudiArabia probably will be largely reduced.**

[REPLY] We agree with the reviewer that finally a combined static and dynamic approach might be a solution for operational setups. However, we selected to perform two totally independent runs in order to clearly demonstrate the use of dynamic NDVI sources as an alternative method

to the static approach for DREAM model without incorporating a vegetation map. We believe that in this way the advances and deficiencies of our development are more evident.

**8. Here only modeled AOD in the two simulations are compared. I wonder if you also see any improvement in other aspects of dust life cycle such as surface dust concentration, vertical distribution, and deposition.**
[REPLY] We focused on AOD for the verification since AOD observations are more regular, available and reliable than observations of profiles, surface concentrations and deposition.

**Minor points:**
**1. Line 28, add "e.g.," before "Torge et al., 2011"**
[REPLY] Done.

**2. Line 31, add space between "precipitation" and "processes". Please fix all similaroccurrences.**
[REPLY] Done.

**3. Line 45, I don't think there is any "feedbacks" on "human health", please considerreorganizing the sentence.**
[REPLY] Indeed, thank you. We have replaced "feedbacks" with "effects".

**4. Line 63, please add brackets for "2013", and fix all similar occurrences.**
[REPLY]Done, thank you.

**5. Line 79, I think the original dust source function developed by Ginoux et al. (2001) did not use "Olson World Ecosystems dataset". Can you explain a bit more here?**
[REPLY] The reviewer is correct. We have corrected this sentence in the revised text.

**6. Line 109, you may want to add a line or two to explain why NDVI of 0.1 is selected instead of 0.15 as used by Kim et al. (2013).**
[REPLY] Indeed it is not easy to define a "best estimate" threshold for all satellite NDVI sensors worldwide. A choice of 0.15 may be more representative on a global base as used by Kim et al. (2013) for AVHRR. Here we adopted the 0.1 threshold based also on previous studies at the area (Solomos et al., 2017) since due to the bareness of the specific modeling domain a higher value could overestimate the dust sources. This discussion is now added in the revised version.

**7. Lines 200-201, not clear. Did you use Ångström exponent to mask AOD? In thatcase, the masked AOD may contain large particles such as dust and sea salt.**
[REPLY] Yes this does not exclude sea salt but the contribution of marine particles to the total AOD is limited.

**8. Fig. 3, it is better to mask out AOD outside the model domain in Fig. 3c for an easy comparison among the three plots.**

[REPLY]Done.

**9. Please clarify in Table 1 caption that this is for annual mean. And for correlation, do you use monthly data? Can you also mark whether the correlation coefficients are statistically significant?**
[REPLY]Done. For correlation we use the daily AERONET data.

**10. Table 2, are monthly or daily data used for correlation? Please add significance test as well.**
[REPLY] We use individual AERONET measurements. The model retrievals are interpolated in time to match the AERONET measurement time. This is now stated more clearly in the revised text. To highlight the significance of correlation, in Table 2 we indicate with bold font all coefficients with p value < 0.01.

**11. Fig. 5, are the time series calculated from single AERONET site (which one?) oraveraged over four stations on the Arabian Peninsula?**
[REPLY]Fig.5 is averaged from the four Arabian stations. This is now more clearly stated in the text.

References:
Ginoux, P., Chin, M., Tegen, I., Prospero, J. M., Holben, B., Dubovik, O., & Lin, S.-J., Sources and distributions of dust aerosols simulated with the GOCART model. J.Geophys. Res., 106(D17), 20, 255–20, 273, https://doi.org/10.1029/2000JD000053,2001
Kim, D., M. Chin, H. Bian, Q. Tan, M. E. Brown, T. Zheng, R. You, T. Diehl, P. Ginoux,and T. Kucsera, The effect of the dynamic surface bareness on dust sourcefunction, emission, and distribution, J. Geophys. Res. Atmos., 118, 871–886, doi:10.1029/2012JD017907, 2013.

---

## Author Response (AR2)

The changes are:

1. Updated Figure_5 with scatter density plots and statistical metrics 2. Table_2 title updated to include : "Statistical metrics from the annual runs (2016) at AERONET stations."